# Impact of Electromagnetic Stirring Roller Arrangement Pattern on Magnetic Field Simulation and Solidification Structure of PW800 Steel in the Second Cooling Zone

**DOI:** 10.3390/ma17051038

**Published:** 2024-02-23

**Authors:** Zhixiang Xiao, Guifang Zhang, Daiwei Liu, Chenhui Wu

**Affiliations:** 1Faculty of Metallurgical and Energy Engineering, Kunming University of Science and Technology, Kunming 650093, China; xiaozhixiang1998@163.com (Z.X.); liudaiwei@pzhu.edu.cn (D.L.); 2Panzhihua Iron and Steel Co., Ltd., Panzhihua 617000, China; wch_neu@126.com

**Keywords:** continuously cast slab, strand electromagnetic stirring, secondary cooling zone, magnetic field, numerical simulation

## Abstract

Strand electromagnetic stirring (S-EMS), a technique applied in the secondary cooling zone, enhances the solidification structure of casting slabs. This study examines how the arrangement pattern of electromagnetic stirring rollers—face-to-face, side-to-side or up-down misalignment produces this enhancement. It uses simulations to analyze the electromagnetic field distribution in these configurations. The findings demonstrate that: (1) The magnetic flux density distribution in the casting slab is related to the arrangement pattern of the electromagnetic stirring rollers. (2) The face-to-face arrangement produces the largest and most concentrated electromagnetic force compared to the other two arrangement patterns. (3) S-EMS can effectively improve the equiaxed grain ratio of casting slabs. Before and after EMS is turned on, casting slabs’ average equiaxed grain ratio goes up from 8% to 33%.

## 1. Introduction

As consumers’ demands for high-quality casting slabs grow, it is essential to improve their quality accordingly. Prior research has demonstrated that the implementation of electromagnetic stirring in continuous casting manufacturing may significantly enhance the quality of casting slabs. This includes augmenting the ratio of equiaxed grain, removing central shrinkage cavities and minimizing the extent of casting slab segregation [1,2]. Due to the high temperature of molten steel, many parameters in the continuous casting process cannot be directly measured, so researchers introduced numerical simulation technology, which is widely used to simulate and predict the flow and temperature fields of the molten steel in the continuous casting process [3,4,5,6,7,8].

Correlation research has been developed in recent decades by some scholars to elucidate the effect of various factors of electromagnetic stirring on S-EMS. For example, Zhang et al. [9] conducted numerical simulations to analyze the electromagnetic field and molten steel flow field in a continuous casting machine under three stirring modes: double butterfly, double ring and three rings. Their findings indicate that near the electromagnetic stirring rolls, the magnetic induction intensity’s distribution is similar across all three modes. However, the electromagnetic force in the double-ring mode is approximately double that of the double-butterfly and three-ring modes. In the longitudinal cross-section at the casting slab’s center, the flow field is symmetrical for all stirring modes, with the double-ring mode generating a larger reflux zone. Li et al. [10] explored the advantages and disadvantages of double-ring and double-butterfly stirring modes under different electromagnetic stirring roller spacings by means of numerical simulation and concluded that the double-ring is better when the spacing of the two sets of electromagnetic stirring roller is five times the roller diameter and the double-butterfly is better when the spacing is 0 [10]. Xiao et al. [11,12] observed a substantial end effect when the electromagnetic stirring roller was installed parallel to the casting slab section, leading to solidification organization asymmetry at both casting slab ends. Based on segmental simulation, they concluded that a 150-mm electromagnetic stirring roller bias achieves near symmetry in solidification. Additionally, they found that increasing the electromagnetic roll pair notably augmented the electromagnetic force in the casting direction and at the roll’s center. Under specific parameters (400 A, 7 Hz), this enhancement increased the equiaxed grain ratio in the casting slab’s center to 69%, facilitating a uniform, dense solidification structure produced, thereby improving hot working properties. Fang et al. [13] conducted a study on the impact of electromagnetic stirring in the second cooling zone of ultrawide casting slabs. The research focused on how different methods of energizing electromagnetic stirring rolls affect the flow of molten steel within the casting slab. Gong et al. [14] observed that varying electromagnetic stirring parameters in their study, notably current intensity, frequency and the stirring mode of electromagnetic stirring rollers, significantly influenced the intensity of stirring and the pattern of steel flow in the casting slab. These factors are intricately linked to the metallurgical efficacy of the stirrer, as they directly impact the temperature distribution within the casting slab and, consequentially, the final solidification structure.

These studies indicate that although lots of research has been done on the effect of the stirring mode, stirring current strength, frequency and electromagnetic stirring roll pairs on S-EMS, the effects of the electromagnetic stirring roller arrangement pattern on the S-EMS are rarely reported. Hence, this study utilizes simulation methods, incorporating the development of a three-dimensional transient magnetic field model to a 1300 mm × 230 mm casting slab to investigate the impact of electromagnetic stirring roller alignments, including face-to-face, side-to-side and up-down misalignment arrangements, on the magnetic field. The research findings offer a reference for optimizing the location of the electromagnetic stirring roller in the secondary cooling zone.

## 2. Model Descriptions

### 2.1. Assumptions

Typically, while performing numerical calculations for electromagnetic fields, certain assumptions are made to simplify the calculations while still guaranteeing that the desired results match the necessary criteria as follows [13,15]:Given the ample space surrounding the electromagnetic stirring device, there is assurance that the magnetic field lines can form a closed loop and are not significantly affected by the external magnetic field. Therefore, it is assumed that the magnetic field in the proximity of the magnetic pole is uniformly distributed, with the magnetic field lines passing through the pole oriented at a right angle to the surface of the pole;The physical parameters are time-invariant constants due to the small variation in the physical parameters of the molten steel during the continuous casting process;Considering the magnetic Reynolds number being less than 1 during electromagnetic stirring in the continuous casting process, it is assumed that the electromagnetic field is barely affected by the slow flow of molten steel (0.9 m/min) [16];The electromagnetic field generated by electromagnetic stirring can be considered a quasistatic field due to the low frequency of the alternating current (5–9 Hz).

### 2.2. Governing Equations

The linear electromagnetic stirring method employs a low-frequency alternating current to exert an electromagnetic force on the sluggish flow of molten steel within the casting slab, enhancing the recirculating flow of the molten metal. The mechanism of this method is shown in the following text and Figure 1. After the application of the alternating current, an alternating magnetic field $\overset{⃑}{B}$ is induced by the electromagnetic stirring roller. This alternating magnetic field $\overset{⃑}{B}$ manifests as a traveling wave magnetic field moving unidirectionally at a specified velocity. Within the casting slab, this traveling wave magnetic field generates induced eddy currents $\overset{⃑}{J}$. Subsequently, the interplay between these induced currents $\overset{⃑}{J}$ and the alternating magnetic field $\overset{⃑}{B}$ results in the emergence of electromagnetic forces $\overset{⃑}{F}$ within the molten steel in the casting slab. For the theoretical modeling of this process, a set of simplified Maxwell’s equations is utilized as delineated in multiple scholarly sources [17,18,19,20].
(1)$$\nabla \times \overset{⃑}{H}=\overset{⃑}{J}+\frac{\partial \vec{D}}{\partial t}\approx \overset{⃑}{J}$$
(2)$$\nabla \times \overset{⃑}{E}=-\frac{\partial \overset{⃑}{B}}{\partial t}$$
(3)$$\nabla \cdot \overset{⃑}{B}=0$$
(4)$$\overset{⃑}{J}=\sigma \overset{⃑}{E}$$
(5)$$\overset{⃑}{F}=\frac{1}{2}Re\left(\overset{⃑}{J}\times \overset{⃑}{B^{x}}\right)$$
where, $\overset{⃑}{H}$ represents the magnetic field intensity, A/m; $\overset{⃑}{J}$ represents the current density, A/m^2^; $\overset{⃑}{E}$ represents the electric field intensity, V/m^2^; $\overset{⃑}{B}$ represents the magnetic induction intensity, mT; $\overset{⃑}{D}$ is the electrical induction intensity, C/m^2^; σ is the conductivity, S/m; *Re* represents the real part of the complex number; $\overset{⃑}{B^{x}}$ represents the conjugate complex of $\overset{⃑}{B}$; and $\overset{⃑}{F}$ represents time-averaged electromagnetic force, N/m^3^.

### 2.3. Electromagnetic Stirring Device

Under different electromagnetic stirring modes, the separation between the centers of the two electromagnetic stirring rollers in the secondary cooling zone is determined to be twice their diameter, equating to a distance of 2 × 240 mm between each roller. The coils are energized through the application of a two-phase alternating current. The formula for computing the current density within a coil energized by an alternating current is expressed as
(6)$$\overset{⃑}{J_{1}}=\overset{⃑}{J_{0}}sin\left(\omega t\right)$$
(7)$$\overset{⃑}{J_{2}}=\overset{⃑}{J_{0}}sin\left(\omega t+90{}^{\circ}\right)$$
where ω represents the angular velocity, $\omega =2\pi f=\frac{2\pi }{T}$, rad/s; f represents frequency, Hz; T represents the period, s; t represents time, s; $\overset{⃑}{J_{0}}$ is the amplitude of the coil current, A/m^2^; and $\overset{⃑}{J_{1}}$ and $\overset{⃑}{J_{2}}$ are the current densities of the first and second phases.

A schematic representation of the electromagnetic stirring roller, along with the power supply methodology, is depicted in Figure 2. It is necessary here to explain simply the role of each part in the picture, as follows:Coil: the coil produces an alternating magnetic field when an alternating current is applied to it;Iron core: the iron core is the carrier of magnetic lines of force, which can enhance and concentrate magnetic fields;Slab: the slab is composed of an externally solidified shell and internal molten steel, with the flow field of the internal molten steel to changes in the electromagnetic force.
Figure 2Electromagnetic stirring roller and its electrification method.
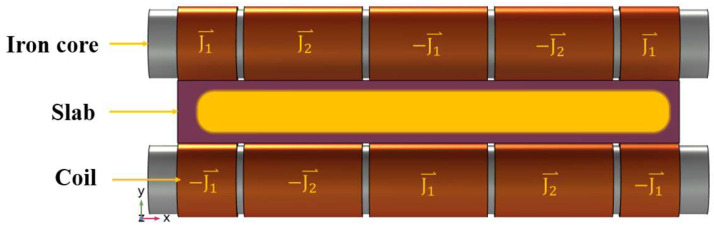



## 3. Results and Discussion

### 3.1. Model Validation

The precision and dependability of the constructed model were verified by conducting a thorough comparison and validation of the data obtained from Xiao et al.’s [12] measurement of the casting slab magnetic field against the results produced by the model in this study. Figure 3 illustrates that the magnetic induction values simulated at the centerline of the electromagnetic stirring roller in the wide direction of the continuous casting slab closely align with those measured in the literature. Specifically, at 0.16 m on the centerline, the peak magnetic induction intensity is noted at 57.9 mT measured and 56.5 mT simulated, with a minor difference of 1.4 mT. The general agreement of measured and simulated values, coupled with a calculation error within acceptable limits, lends credibility to the numerical simulation results.

### 3.2. Selection of Optimal Stirring Current and Frequency

Based on the rated current and frequency parameters of an electromagnetic stirrer in a specific steel plant, where a pair of electromagnetic stirring rollers were arranged in a face-to-face configuration, two sets of simulations were conducted. In the initial series of simulations, the stirring frequency was uniformly established at 7 Hz, while the stirring currents were assigned values of 340 A, 360 A, 380 A and 400 A, respectively. The optimal stirring current was determined based on the magnitude of the generated electromagnetic force. In the subsequent series of simulations, the stirring current was uniformly established as the optimal value identified from the first series, with the stirring frequencies being set at 5 Hz, 6 Hz, 7 Hz, 8 Hz and 9 Hz, respectively. The optimal stirring frequency was also determined based on the magnitude of the electromagnetic force generated.

Figure 4 demonstrates that the greatest electromagnetic force generated by the electromagnetic stirring roller occurs when the current is set at 400 A. Given the minimal disparity in electromagnetic force between 400 A and 380 A, and considering that excessive electromagnetic force may deteriorate casting slab segregation and impair the solidification structure, 380 A has been selected as the optimal stirring current. Excessive electromagnetic force may lead to the breakthrough of the solidification shell by electromagnetic force-driven molten steel flow, resulting in molten steel leakage. Additionally, elevated frequencies contribute to increased energy loss through the skin effect. Taking a conservative approach, the optimal frequency is determined to be 5 Hz.

### 3.3. Distribution of Electromagnetic Force and Magnetic Flux Density When Two Pairs of Electromagnetic Stirring Rollers Are Arranged Face-to-Face

The geometric schematic diagram depicting the two pairs of electromagnetic stirring rollers arranged face-to-face is presented in Figure 5a. Figure 5b presents the magnetic flux density distribution in the longitudinal section of the casting slab center at a current of 380 A, a frequency of 5 Hz and during a period of 5/8 T. The presence of the para-position electromagnetic stirring roller causes the magnetic field to close at the coil’s corresponding position inside the casting slab, resulting in a localized peak in magnetic flux density where the coil is situated.

Figure 5c depicts the distribution of electromagnetic force in the central longitudinal section of the casting slab. The concentration of the magnetic field generated by the coil at this position, interacting with the molten steel in the casting slab, results in an increased local electromagnetic force. So, the electromagnetic force at the location of the electromagnetic stirring roller is observed to be higher than that in the surrounding areas, with a peak in electromagnetic force manifesting at the coil position of the roller.

### 3.4. Distribution of Electromagnetic Force and Magnetic Flux Density When Two Pairs of Electromagnetic Stirring Rollers Are Arranged Side-to-Side

The geometric schematic diagram depicting the two pairs of electromagnetic stirring rollers arranged side-to-side is presented in Figure 6a. Figure 6b presents the magnetic flux density distribution in the longitudinal section of the casting slab center at a current of 380 A, a frequency of 5 Hz and during a period of 5/8 T. When electromagnetic stirring rollers are arranged side-to-side, the lack of opposing rollers leads to the magnetic flux density vector being oriented predominantly in a single direction. This specific orientation results in the magnetic induction lines passing through the casting slab at a certain angle and closing at the external electromagnetic stirring rollers. Consequently, this results in a higher magnetic flux density near the rollers compared to a face-to-face arrangement, with the exception of the coil position.

Figure 6c depicts the distribution of electromagnetic force in the central longitudinal section of the casting slab. Besides the component force aligned with the direction of magnetic field movement, significant upward and downward components of electromagnetic force are also present. The mechanism entails the generation of a magnetic field through the side-to-side arrangement of electromagnetic stirring rollers, resulting in the formation of a curved magnetic induction line. It exhibits a semicircular distribution within the casting slab (referring to Figure 1). As the molten steel within the casting slab flows, it cuts through the magnetic induction line that is not perpendicular to the wide side of the casting slab, generating a significant electromagnetic component force.

### 3.5. Distribution of Electromagnetic Force and Magnetic Flux Density When Four Electromagnetic Stirring Rollers Are Arranged in an Up-Down Misalignment

The geometric schematic diagram depicting the two pairs of electromagnetic stirring rollers arranged in an up-down misalignment is presented in Figure 7a. Figure 7b presents the magnetic flux density distribution in the longitudinal section of the casting slab center at a current of 380 A, a frequency of 5 Hz and during a period of 5/8 T. In the up-down misalignment arrangement, the flux density is higher near the electromagnetic stirring rollers and the peak flux density happens at the coil position, similar to what was seen in the side-to-side arrangement. However, this increased flux density is more pronounced in the up-down misalignment than in the side-to-side configuration. Furthermore, a pair of oppositely misaligned rollers leads to the closure of the magnetic field within the casting slab at a distinct tilt angle. This tilt angle is determined by the angle between the oppositely misaligned rollers and the casting slab’s wide surface, resulting in the peak magnetic flux density area intersecting near the closest misaligned roller.

Figure 7c depicts the distribution of electromagnetic force in the central longitudinal section of the casting slab. From the figure, it can be seen that the overall electromagnetic force is higher when the electromagnetic stirring rollers are arranged in an up-down misalignment compared to when they are arranged side-to-side. There is also the electromagnetic component force when the electromagnetic stirring rollers are arranged in an up-down misalignment and the mechanism of generating this component is the same as that of generating electromagnetic component force when they are arranged side-to-side.

The study selected the centerline in different directions as the data value line to compare the distribution of electromagnetic force under different arrangement patterns of the electromagnetic stirring roller. Figure 8 displays the results. At a cycle of 1/2 T, the observed maximum electromagnetic force on the casting slab’s centerline for the face-to-face, side-to-side and up-down misalignment arrangements is 2378 N/m^3^, 31607 N/m^3^ and 1516 N/m^3^, respectively. The corresponding average electromagnetic forces for these arrangements are 432 N/m^3^, 646 N/m^3^, and 604 N/m^3^. In addition, the maximum electromagnetic force exerted on the wide centerline of the electromagnetic stirring roller casting slab reaches 9850 N/m^3^, 4779 N/m^3^ and 4629 N/m^3^, respectively. The average electromagnetic force measures 4712 N/m^3^, 2766 N/m^3^ and 2696 N/m^3^, respectively. It is evident that face-to-face arrangements have the smallest component of electromagnetic force, suggesting that this arrangement generates the largest and most concentrated electromagnetic force.

### 3.6. The Impact of Electromagnetic Field on PW800 Steel’s Solidification Structure

In the S-EMS industrial experiment, two pairs of electromagnetic stirring rollers were selected for the face-to-face arrangement of PW800 steel casting slab continuous casting. PW800 steel belongs to the category of electrical steel and its chemical composition is presented in Table 1.

The electromagnetic stirring rollers are arranged in the following ways: face-to-face, side-to-side and in an up-down misalignment. Among them, the face-to-face arrangement generates the largest electromagnetic force and the smallest area of effect on the casting slab. This phenomenon accelerates the erosion speed of molten steel at the solidification front and is conducive to the formation of equiaxed grains.

As shown in Figure 9a,b, the solidification structures of casting slabs, produced with or without S-EMS, were compared in industrial experiments. Figure 9c illustrates a schematic diagram for calculating the equiaxed grain ratio. The grain aspect ratio less than or equal to 3 was defined as an equiaxed grain and the zone where equiaxed grains exist is the equiaxed grain zone. According to the measured thickness of the equiaxed grain zone and Equation (8), the equiaxed grain ratio at 1/4, 2/4 (center) and 3/4 of the casting slab width was calculated and the average value was taken to obtain the equiaxed grain ratio of the casting slab.
(8)$$\Phi =\frac{H_{e}}{H}\times 100$$
where Φ is the equiaxed grain ratio, %; $H_{e}$ is the thickness of the equiaxed grain zone, mm; and H is the thickness of the casting slab, constant, 230 mm.

In the absence of S-EMS, the heat transfer from the interior of the casting slab to the surface transpires at a more gradual pace, leading to a slower dissipation of superheat. Superheat is defined as the difference between the actual temperature of the molten steel and its liquidus temperature. As long as the molten steel remains superheated above the temperature of the liquid phase line, it solidifies in the form of dendritic grains growing along the direction of the temperature gradient. Consequently, the macrostructure of the casting slab reveals more developed columnar grains [21,22]. During the solidification process, the interdendritic liquid enriched in alloying elements migrates to the center before solidification. Because the center of the liquid pool lacks equiaxed grains to decentralize the enriched liquid, the flow of the residual enriched liquid will gather in the center of the slab. This led to serious center segregation after the solidification process was completed [21,23]. When using S-EMS with 380 A and 5 Hz electromagnetic stirring parameters, on the one hand, the electromagnetic force generated by the traveling wave magnetic field causes the molten steel to flow violently and wash away the solidification front, which homogenizes the temperature at the solidification front of the casting slab and accelerates the dissipation of superheat. This suppresses the growth of columnar grains and promotes the formation of equiaxed grains [21,24,25]. On the other hand, under the action of electromagnetic force-driven molten steel flow washing, the tips of columnar grains are remelted or broken. The growth of columnar grains is inhibited and broken tips of columnar grains are washed into the molten steel, serving as nuclei for equiaxed grain nucleation and promoting the formation of equiaxed grains [26]. The thickness of the equiaxed grain zone $H_{e}$ at 1/4, 2/4 (center) and 3/4 of the casting slab width measured from Figure 9a are 16 mm, 21 mm and 18 mm. From Equation (8), the equiaxed grain ratios are calculated as 7%, 9% and 8%, respectively. The average equiaxed grain ratio of the casting slab is 8%.

Upon observation of the macroscopic structure of the casting slab following the application of S-EMS, two distinct white light bands are clearly visible as a result of electromagnetic stirring. Additionally, the amount of equiaxed grains in the casting slab is noticeably enhanced. The thickness of the equiaxed grain zone $H_{e}$ at 1/4, 2/4 (center) and 3/4 positions of the casting slab width exhibit an increase to 78 mm, 81 mm and 69 mm. From Equation (8), the equiaxed grain ratios Φ are calculated as 34%, 35% and 30%, respectively. The average ratio of equiaxed grain in the casting slab ranged from 8% to 33%. However, the resulting grain structure distribution is unequal, with a concentration of equiaxed grains on one side while the other side is predominantly composed of columnar grains. This effect is caused by the arc-shaped construction of the continuous casting machine. The curve in the curved structural segment of the continuous casting machine (usually mold and secondary cooling zone) has a certain curvature. The casting slab is accordingly divided into two sides: the loose side and fixed side. The term “loose side” denotes the inner arc of the casting slab, whereas the term “fixed side” designates the outer arc of the casting slab. Because the loose side surface of the casting slab is a water accumulation surface, the cooling speed on the loose side is faster than the cooling speed on the fixed side during the water spray cooling process, promoting the formation of columnar grains on the loose side. This accelerates the formation of columnar grains on the loose side. At the same time, the primary grain nucleus deposits at the solidification front of the fixed side due to gravity, impeding the formation of columnar grains. The amalgamation of these two events leads to an irregular distribution of grain structure within the casting slab [27].

After the application of S-EMS, the equiaxed grain ratio of the casting slab is significantly increased. Previous studies have shown that, for electrical steels, a higher equiaxed grain ratio is closely related to better overall mechanical and magnetic properties [28].

## 4. Conclusions

The distribution of magnetic flux density in the casting slab is determined by the arrangement pattern of the electromagnetic stirring roller. In face-to-face, side-to-side and up-down misalignment arrangements, peak magnetic flux density occurs at the coil position. Furthermore, the flux density adjacent to the electromagnetic stirring roller in the side-to-side and up-down misalignment arrangements is observed to be higher than in the face-to-face arrangement.The electromagnetic force of the face-to-face arrangement generated by the largest and most concentrated produces almost no upward and downward components of electromagnetic force. Therefore, the face-to-face arrangement is superior to the remaining two types of arrangement patterns. The arrangement of side-to-side and up-down misalignment will produce large upward or downward components of electromagnetic force. These component forces lead to a reduction in the force applied to the stirring process and could potentially disturb the vortex, thereby compromising the effectiveness of the stirring.Using S-EMS significantly enhances the equiaxed grain ratio of casting slabs. Before and after the use of S-EMS, the equiaxial grain thickness of casting slabs increased from 16 mm, 21 mm and 18 mm to 78 mm, 81 mm, and 69 mm and the equiaxed grain ratio of casting slabs increased from 7%, 9% and 8% to 34%, 35% and 30%, respectively. The average equiaxed grain ratio of the casting slabs increased from 8% to 33%. Additionally, in actual production, the presence of curvature in the casting slabs leads to nonuniform heat transfer, causing the growth of columnar grains to be promoted on the loose side while hindering their growth on the fixed side.

## Figures and Tables

**Figure 1 materials-17-01038-f001:**
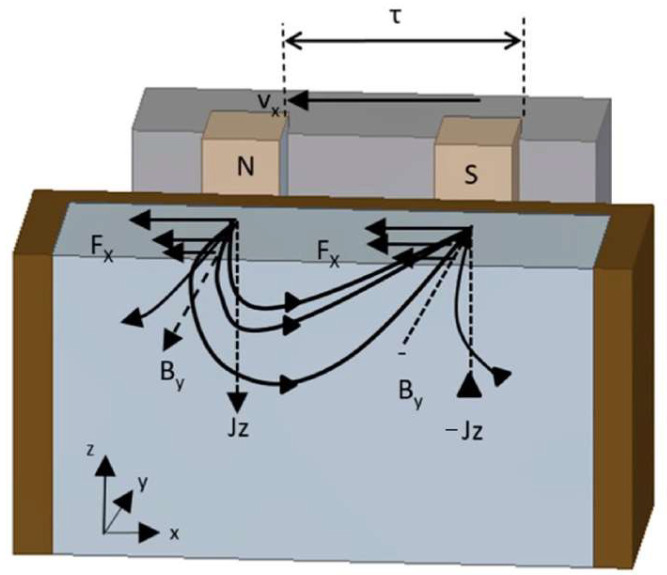
Working principle of the linear stirrer [12].

**Figure 3 materials-17-01038-f003:**
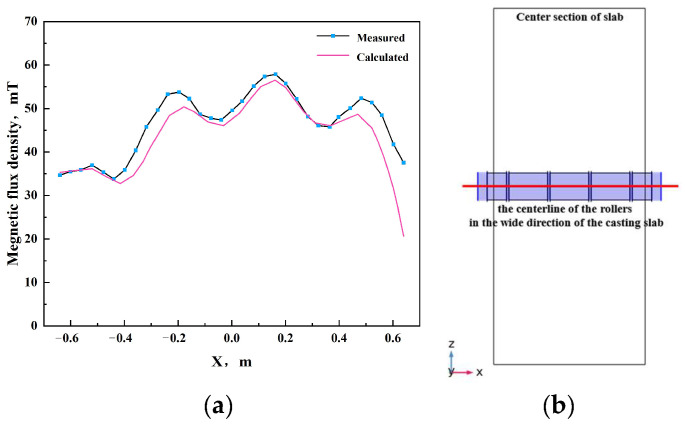
(**a**) Distribution of magnetic field density along the centerline of the electromagnetic stirring roller in the wide direction of the casting slab under electromagnetic stirring. (**b**) The centerline of the rollers in the wide direction of the casting slab schema.

**Figure 4 materials-17-01038-f004:**
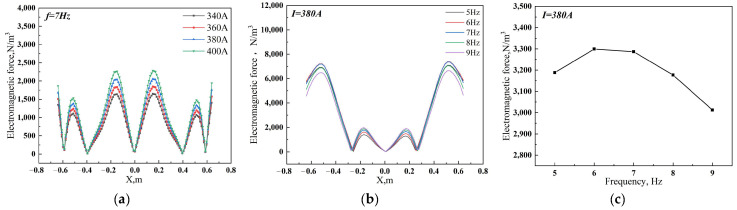
Distribution of electromagnetic force along the centerline of the rollers in the casting slab wide direction: (**a**) different currents; (**b**) different frequencies; (**c**) the average electromagnetic force at different frequencies.

**Figure 5 materials-17-01038-f005:**
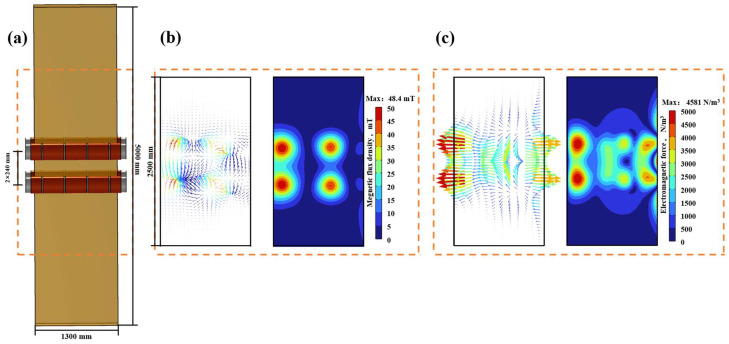
Distribution of magnetic induction intensity inside continuous casting slab under electromagnetic stirring arranged face-to-face: (**a**) geometric schematic diagram; (**b**) magnetic flux density cloud diagram and vector diagram; (**c**) electromagnetic force cloud diagram and vector diagram.

**Figure 6 materials-17-01038-f006:**
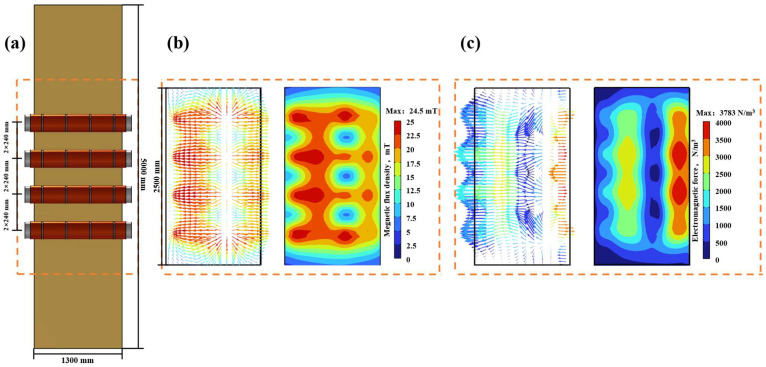
Distribution of magnetic flux density and electromagnetic force in the casting slab with two pairs of electromagnetic stirring rollers arranged side-to-side: (**a**) geometric schematic diagram; (**b**) magnetic flux density cloud diagram and vector diagram; (**c**) electromagnetic force cloud diagram and vector diagram.

**Figure 7 materials-17-01038-f007:**
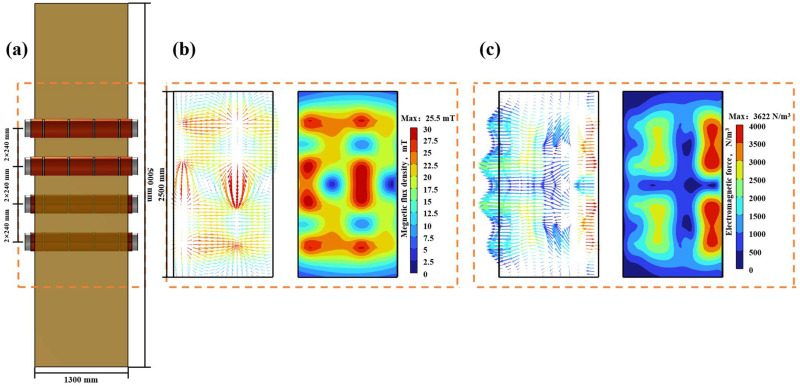
The distribution of magnetic flux density and electromagnetic force in the casting slab with two pairs of electromagnetic stirring rollers arranged in an up-down misalignment: (**a**) geometric schematic diagram; (**b**) magnetic flux density cloud diagram and vector diagram; (**c**) electromagnetic force cloud diagram and vector diagram.

**Figure 8 materials-17-01038-f008:**
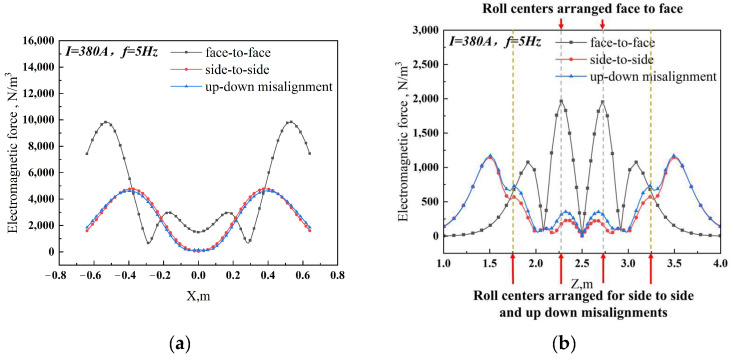
Distribution of electromagnetic force with different arrangement patterns (**a**) along the centerline of the upper pair of rollers of face-to-face arrangement in the casting slab wide direction; (**b**) along the centerline in the casting direction.

**Figure 9 materials-17-01038-f009:**
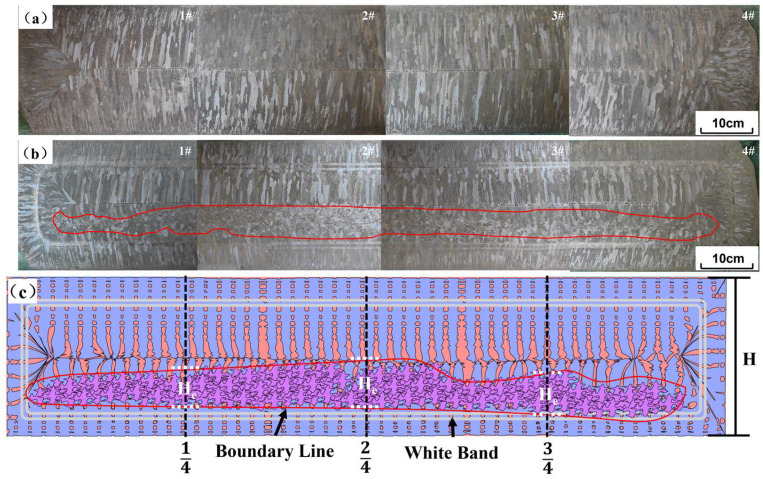
Cross-section macrostructure photos of continuous casting slabs and a schematic diagram of equiaxed grain ratio calculation: (**a**) without S-EMS; (**b**) with S-EMS (face-to-face); (**c**) schematic diagram for calculating the equiaxed grain ratio of the cross-section of the casting slab.

**Table 1 materials-17-01038-t001:** Chemical compositions of PW800 steel (wt.%).

C	Si	Mn	P	S	Al	Cr	Cu	Ni
0.003	1.05	0.32	0.012	0.005	0.31	0.0285	0.041	0.033

## Data Availability

The data presented in this study are available on request from the corresponding author due to the project policy.

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
