# Peer review of "Impact of Electromagnetic Stirring Roller Arrangement Pattern on Magnetic Field Simulation and Solidification Structure of PW800 Steel in the Second Cooling Zone"

_materials, 2024, doi:10.3390/ma17051038_

Round 1
Reviewer 1 Report
Comments and Suggestions for Authors
Authors have described strand Electromagnetic Stirring technique by rollers with face-to-face, side-to-side or up-down misalignments to produce enhancement of the solidification structure of slabs. . There are several technical questions which needs to be answered before this paper can be accepted for publication. Please see the annotated pdf file for the detailed comments from reviewer. Thank you.

Please see the attached pdf file for minor English correction, especially grammars.
Reviewer 2 Report
Comments and Suggestions for Authors
referee report
materials-2816424-peer-review-v1
Impact of Electromagnetic Stirring Roller Arrangement Pattern on Magnetic Field Simulation and Solidification Structure of
PW800 Steel in the Second Cooling Zone
Zhixiang Xiao et al.
The present manuscript reports on a thorough analysis of the electromagnetic stirring technique enhancing the solidification structure of slabs. This is an
interesting approach as this situation was rarely studied in the literature. The authors present calculations of the magnetic flux density and the electromagnetic forces,
and Fig. 8 shows the macrostructures of cast slabs together with a calculated grain rate of the cross-section.
The topic is well suited for the Special Issue of Materials.
The present manuscript comprises 8 figures, no table and 21 references are given.
All figures seem to be well prepared but have too low resolution.
The discussion itself is well written and explains the important features. Figure 8 gives a comparison of real cast slabs with and
without S-EMS applied and an analysis. However, the authors do not provide an explanation how fig. 8 (c) was created. Furthermore,
there is no comparison of the resulting slab structures when applying the calculated situations of Figs. 4--6. Supposedly, this is
a must to be done in the future.
In summary, several explanations concerning the modelling itself are missing, Fig. 8(c) needs to be explained properly and some more
comments should be made concerning the resulting properties of the cast slabs.
Some more points concerning the reference list:
# The journal style should be closely followed.
# Ref. [3] is cited completely wrong.
# Ref. [5]: Spelling and writing autors' names should be uniform.
# Ref. [8] is incomplete
# Ref. [9]: What means 335-340+350 ?!
Thus, the manuscript may be published after a major revision.
Comments on the Quality of English LanguageExtensive improvement required
Reviewer 3 Report
Comments and Suggestions for Authors
Dear Authors
I read the results of your research with interest.
I believe that the text requires the following additions.
1. It should be indicated what impact the assumptions/simplifications may have on the model results and its limitations.
2. The limitations of the test/simulation resulting from the adopted geometric limitation should be indicated.
3. The choice of PW800 steel grade should be justified and its chemical composition for which the simulations were carried out should be presented.
Reviewer 4 Report
Comments and Suggestions for Authors
After a thorough review of your paper, “Impact of Electromagnetic Stirring Roller Arrangement Pattern on Magnetic Field Simulation and Solidification Structure of PW800 Steel in the Second Cooling Zone”, I am convinced that your work should be accepted for publication. Your conclusions further strengthen this belief:
You have clearly demonstrated that the distribution of magnetic flux density in the slab is determined by the arrangement pattern of the electromagnetic stirring roller. This is a significant finding that adds depth to our understanding of the process.
Your finding that the face-to-face arrangement generates the largest and most concentrated electromagnetic force with minimal upward and downward components is particularly noteworthy. This insight could have important implications for the design and operation of electromagnetic stirring systems.
The significant enhancement in the equiaxial crystallization rate of slabs with the use of S-EMS is a compelling result. The increase from 8% to 33% is substantial and underscores the effectiveness of S-EMS.
Your paper is well-written, logically organized, and presents a robust methodology. The findings are significant and contribute to the field. Congratulations on your excellent work.
Round 2
Reviewer 2 Report
Comments and Suggestions for Authors
Well performed revision. The manuscript is now suitable for publication.
Comments on the Quality of English LanguageNow acceptable